# Characterization of the *TcCYPE2* Gene and Its Role in Regulating Trehalose Metabolism in Response to High CO$_2$ Stress

**Yan-Fei Zhou** [1,†], **Min Zhou** [2,†], **Yuan-Yuan Wang** [2], **Xin-Yi Jiang** [1], **Pei Zhang** [3], **Kang-Kang Xu** [2], **Bin Tang** [1,*] and **Can Li** [2,*]





[1] College of Life and Environmental Sciences, Hangzhou Normal University, Hangzhou 311121, China; 19858108489@163.com (Y.-F.Z.); j1813944830@163.com (X.-Y.J.)
[2] Key Laboratory of Surveillance and Management of Invasive Alien Species in Guizhou Education Department, Guizhou Provincial Key Laboratory for Rare Animal and Economic Insect of the Mountainous Region, College of Biological and Environmental Engineering, Guiyang University, Guiyang 550005, China; zhoumin810611@126.com (M.Z.); wyy5211234@163.com (Y.-Y.W.); kkxu1988@163.com (K.-K.X.)
[3] Jing Hengyi School of Education, Hangzhou Normal University, Hangzhou 311121, China; zhangpei2020@stu.hznu.edu.cn
[*] Correspondence: tbzm611@hznu.edu.cn (B.T.); lican790108@163.com (C.L.)
[†] These authors contributed equally to this work.

**Abstract:** Cytochrome P450 monooxygenase (CYP) is one of the three detoxification metabolic enzymes in insects, and is involved in the metabolism and transformation of endogenous substances as well as the activation and degradation of exogenous compounds. This study aims to reveal the molecular mechanism of *CYP9E2* in *Tribolium castaneum* in adapting to high-CO$_2$ stress. By predicting the sequence function of *CYP9E2*, analyzing the temporal and spatial expression profile of *TcCYP9E2*, and using RNAi to silence *TcCYP9E2* combined with a high-CO$_2$ stress treatment, we measured the carbohydrate content, trehalase activity, and gene expression levels in trehalose metabolism of *T. castaneum*. A bioinformatics analysis showed that the predicted molecular weight of the protein encoded by *TcCYP9E2* is 60.15, the theoretical isoelectric point is 8.63, there is no signal peptide, and the protein is hydrophilic. An evolutionary tree analysis showed that *TcCYP9E2* belongs to the CYP6 family and belongs to the CYP3 group; and the spatiotemporal expression profile results showed that *TcCYP9E2* was highly expressed in the larvae midgut 48 h after injection of dsCYP9E2, with survival rates decreasing with the increase in CO$_2$ concentration. Under the condition of 75% CO$_2$, the contents of glycogen, glucose, ATP, and membrane-bound trehalase decreased significantly after the injection of dsCYP9E2. The expression of *TRE-1*, *TRE-2*, and *GP* in trehalose metabolism and energy pathways was significantly downregulated.

**Keywords:** *Tribolium castaneum*; Cytochrome P450 monooxygenase; High-CO$_2$ treatment; RNAi

## 1. Introduction

Grain storage is an important link in the grain supply, and the safe storage of grain is crucial to the national economy and people's livelihood, which is of great significance for national stability and development [1]. The ideal control conditions can be achieved through measures such as ventilation and cooling to address the issues of heat and mold in stored grain, but it is difficult to avoid the activities of stored pests causing huge losses to stored grains [2]. The use of chemical fumigants such as phosphine and bromomethane to control pests are associated with the problem of pesticide residues, and long-term use induces certain drug resistances in insects [3,4]; the same is true for semiochemicals, which have been widely studied for pest management applications [5,6]. Scientists are beginning to explore environmentally friendly and energy-saving new control methods, with controlled atmosphere technology being one of them. Controlled atmosphere technology is the

process of filling a closed environment with $CO_2$ or $N_2$ or placing a controlled atmosphere agent to regulate the oxygen content in a closed storage environment, forming a low-oxygen state that inhibits the aerobic respiration of pests during storage, affecting their growth, development, and reproduction, or causing death [7–10].

However, long-term exposure to $CO_2$ is not conducive to the storage of rice because it can lead to a decrease in protein content and quality. Moreover, the long-term use of high-$CO_2$ gas or high-$N_2$ gas can lead to stress resistance in pests under low-oxygen conditions, where pests gradually adapt to lower oxygen levels by regulating their physiological functions or changing gene expression levels in their bodies [8,11,12]. Insects produce a series of physiological reactions in hypoxic environments to increase the ventilation around their bodies by compressing their bodies to promote air flow, increasing ventilation time, and reducing molting times to resist hypoxic environments [13]. *Drosophila melanogaster* can improve its adaptability to hypoxic conditions by changing the diameter of the trachea and increasing oxygen intake [14]. *Callosobruchus maculatus* passes through the hypoxia period by slowing down its own growth and development and reducing fecundity [15]. The best insecticidal effect cannot be achieved using a $CO_2$-modified atmosphere alone. Thus, it is necessary to explore more combinations of joint pest control, such as controlled atmospheres, RNA insecticides [16–18], essential oils [19], Diallyl trisulfide (DAT) [20,21] or the combination of these with biological pesticides.

Relevant studies have found that a high-$CO_2$ stress treatment had a significant effect on the biochemical function of detoxification enzymes in stored insects [22]. Cytochrome P450 monooxygenase (CYP) is a supergene family enzyme system that widely exists in eukaryotes [23–25]. It not only has metabolic effects on exogenous substances such as pesticides, plant secondary substances, and other environmental toxic substances, but also participates in some endogenous substances that play important physiological functions such as the hormone metabolism of steroids and fatty acids [26–28]. Among the four gene clusters of the CYP gene family, CYP2 and mito genes are mainly involved in the growth and development of insects, such as the biosynthesis of ecdysteroids, etc. CYP3 and CYP4 genes are involved in responding to external environmental stimuli and are vulnerable to the selection pressure of toxic substances in the external environment, such as pesticides and host metabolites [29,30]. The CYP3 cluster includes two major families: CYP6 and CYP9 [31]. The CYP6 family is unique to the class Insecta and typically participates in the metabolism of insect insecticides and plant secondary metabolites [31–33]. CYP, due to its genetic diversity, broad-spectrum substrate, and complex catalytic reactions, can mediate insect resistance to various insecticides [34,35]. Insects can regulate their defense state by increasing the enzyme activity of CYP and the expression level of CYP genes in order to resist toxic compounds or environments. At present, the involvement of CYP in the detoxification of endogenous metabolites under hypoxic stress has only been reported in *Caenorhabditis elegans* and human tumor cells [36,37]. However, the specific CYP genes and their regulatory mechanisms involved in regulating the endogenous detoxification of storage pests and insects under hypoxic stress have not been reported. Therefore, using CYP as a target for the development of new nucleic acid pesticides under hypoxic stress is highly practical and opportune.

In this paper, *Tribolium castaneum* (Herbst) (Coleoptera: Tenebrionidae), an important grain storage pest, was taken as the research object. It has a fast speed of reproduction and directly harms grain, mainly by eating storage materials, and also secretes malodor, which leads to grain mildew; thus, it causes serious harm to the environment, ecology, and human security [38,39]. Effective prevention and control of *T. castaneum* will help us to avoid the loss of "Ten thousand acres of fertile land", which has far-reaching significance for the safe storage of grain. In recent years, the combination of multiple pesticides and multiple control methods has become a research hotspot for the green prevention and control of stored grain pests. The function of the CYP gene of *T. castaneum* was studied using RNAi technology, combined with the biological and physiological changes caused by high-$CO_2$ stress. This paper explored the potential molecular target *TcCYP9E2* (Gene ID: 103314135), which can

be used for new nucleic acid preparations. Our aim is to reveal the molecular mechanism of CYP in the adaptation of high-$CO_2$ stress to stored grain pests and to provide a theoretical basis for the use of new nucleic acid pesticides in low-oxygen storage technology, providing new ideas and targets for the sustained and effective prevention and control of stored grain pests.

## 2. Materials and Methods

### 2.1. Insect Source and Feeding Method

The tested insects *T. castaneum* were reared in the laboratory of Hangzhou Normal University (Hangzhou, China), raised on whole-wheat flour containing 5% yeast in an incubator at 28 ± 2 °C, 65% relative humidity, and a 0L:24D photoperiod. The 8th instar larvae of *T. castaneum* were taken as test objects.

### 2.2. Bioinformatics Analysis

After using the NCBI (http://www.ncbi.nlm.gov (accessed on 10 March 2021) BLAST webpage target gene comparison, we utilized HMMER (https://www.ebi.ac.uk/Tools/hmmer/ (accessed on 11 March 2021) to predict amino acid domains. The online analysis software ExPASy ProtParam was used to analyze the physical and chemical properties of the encoded protein (amino acid composition, protein-predicted molecular weight KDa, isoelectric point PI, hydrophilicity, etc.). Analysis of the signal peptide of the sequence was performed using SignalP4.1 Server online software to determine the C score, Y score, S score, and mean S-score; the analysis of transmembrane structures was performed using the TMHMM Server v.2.0 online website. The phosphorylation sites were analyzed using the NetPhos 3.1 Server, specifically the threonine (Thr = T) phosphorylation site, serine (Ser = S) phosphorylation site, and tyrosine (Tyr = Y) phosphorylation site. We selected the amino acid sequences of known CYPs. MEGA 11 software was used to construct a phylogenetic tree using the adjacency method, and for bootstrap analysis. We used 1000 replicates to verify the robustness of each cluster and graphically depicted the evolutionary tree via iTOL: Interactive Tree of Life (assembly. de).

### 2.3. Collection of Tissue and Developmental Expression Samples

The development expression samples were collected from different development stages of *T. castaneum* (larvae at 1–8 instars, pupae at 1–4 days, adults at 1 day, adults at 5 days, adults at 10 days, adults at 15 days, adults at 21 days, adults at 30 days). Three biological replicates were set for each development stage, and 35 test insects were collected for each repetition. The tissue expression samples were collected from the midgut, head, epidermis, fat body, Malpighian tubule system, and wing of *T. castaneum*. One hundred heads of 8th instar larvae of *T. castaneum* were obtained, after which their tissues were separated under a Leica EZ4 HD stereomicroscope and stored in THE Ambion® RNA Storage Solution (Thermo Fisher, Waltham, MA, USA, Cat AM7000). Three biological replicates were set up for the experiment, and 100 larvae were anatomized for each repetition. The samples were stored in an ultralow temperature refrigerator at −80 °C on standby for the later detection of the change in CYP9E2 expression level.

### 2.4. Total RNA Extraction and dsRNA Synthesis

The total RNA was extracted from the aforementioned test insect samples using the Trizol reagent kit (Invitrogen, Carlsbad, CA, USA, CAS No. 108-95-2). The RNA extraction solution was sub packed, 1 μL RNA was taken to test the purity and concentration of the extraction, and 2 μL RNA was subjected to 1% agarose gel electrophoresis to check whether the RNA was complete. A PrimeScript® RT reagent kit with the gDNA Eraser reverse transcription kit (TaKaRa, Kyoto, Japan) was used to synthesize the first-strand cDNA.

Afterward, agarose gel electrophoresis was used to detect the target fragment. After cutting the correct band with the target gene, a MiniBEST Agarose Gel DNA Extraction Kit Ver. 4.0 (Takara, Kyoto, Japan) was used for gel recovery. We used a NanoDrop 2000

(Thermo Fisher Scientific, Waltham, MA, USA) to detect quality and concentration and stored the recovered product DNA at $-20$ °C. Thereafter, 3 µL of recovered product, 3.5 µL of Solution I reagent, and 0.5 µL of PMD18-T reagent were added to the PCR tube, followed by micro-centrifugation and incubation in a water bath at 16 °C for 30 min to obtain the connecting solution. Utilizing DH5 $\alpha$ receptive cells, plasmid transformation was carried out, and the evenly growing colonies with smooth edges and transparent colors on the Petri dish were picked and dissolved in 30 µL of sterilized water; then, they were used as a colony PCR template in an PCR tube. After PCR amplification, 1% agarose gel electrophoresis was used to detect the correctness of the target fragment of the colony amplified using PCR. Positive clones were cultured overnight in LB liquid medium (containing Amp) at 37 °C at 250 rpm on a shaking table, and then 500 µL of clones were taken and sent to Zhejiang Sunya Biotech Co., Ltd. (Zhejiang, China) for sequencing. The primers were designed using Primer Premier 6.0 software with a T7 promoter to perform cross-PCR reaction on the correctly sequenced plasmid, after which the dsRNA of the target gene was synthesized according to the instructions of the T7 RiboMAX™ Express RNAi System (Promega, Madison, WI, USA, REF P1700). The concentration of the synthesized dsRNA was determined using a NanoDrop 2000 (Thermo Fisher Scientific, Waltham, MA, USA), and the dsRNA of green fluorescent protein gene (GFP) was synthesized via the same method, with the GFP gene as the control group.

### 2.5. Microinjection of dsRNA

The 8th instar larvae of *T. castaneum* were selected and placed on ice for paralysis. Using a Transferman 4r microinjector (Eppendorf, Hamburg, German), dsCYP9E2 (100 nL, 2000 ng/µL) was injected into the abdomen between the second and third segments of the backs of the *T. castaneum* larvae, and the same amount of dsGFP was injected for the control group. We placed the treated larvae in a feeding bottle and set up air groups and $CO_2$ groups with different concentrations (25% $CO_2$ + 75% air; 50% $CO_2$ + 50% air; 75% $CO_2$ + 25% air; and 95% $CO_2$ + 5% air). Each group had three replicates, with 40 larvae per replicate.

### 2.6. Determination of Trehalase Activity, Carbohydrate Content, and ATP Content

The detection of trehalase activity was slightly modified according to Tatun's method [40,41]. For each treatment, 60 larvae of *T. castaneum* were utilized for each of 3 biological replicates. First, 200 µLPBS of ice was added to the homogenated sample and the sample was sonicated for 30 min; later on, 800 µL of PBS was added to the crushed sample at 4 °C, $1000\times$ $g$, and centrifuged the sample for 20 min. After centrifugation, 350 µL of supernatant was used to determine protein concentration, glycogen, and trehalose content, and 350 µL supernatant underwent ultracentrifugation at $20{,}800\times$ $g$ for 60 min. After ultracentrifugation, 300 µL of supernatant was used to determine glucose concentration, trehalase activity, and protein content. We suspended the supercentrifuge precipitation in 300 µL of added PBS to determine glucose concentration, trehalase activity, and protein concentration. Then, we combined a mixture of the supernatant and suspension (60 µL), 75 µL of 40 mM trehalose (Sigma Aldrich, Saint Louis, MO, USA, CAS 99-20-7), and 165 µL of PBS, which was incubated at 37 °C for 60 min and inactivated by 5 min incubation at 100 °C. The trehalase activity was measured using the glucose (Go) Assay Kit (Sigma, MO, USA), and the reaction was terminated by adding 12 N $H_2SO_4$ (260 µL, CAS 7664-93-9). Finally, the absorbance value was measured at 540 nm using a microplate reader (Thermo Fisher Scientific, Waltham, MA, USA). The protein contents of samples were determined using the BCA Protein Assay Kit (Beyotime, Shanghai, China).

Firstly, *T. castaneum* samples were washed with normal saline, after which double-distilled water was added to prepare a 10% tissue homogenate of *T. castaneum*. Then, the samples were placed in a boiling water bath for 10 min. The samples were removed and placed on a vortex oscillator to fully mix for 1 min, 3500 r/min, after which they were centrifuged for 10 min. The supernatant was extracted, and the ATP content

was calculated by measuring the absorbance value at 636 nm according to the instructions of the ATP content testing kit (Nanjing Jiancheng Bioengineering, Nanjing, China, Item number: A095-1-1).

*2.7. qRT-PCR*

The survival of each group of test insects after feeding them with dsRNA combined with $CO_2$ was observed and recorded 48 h after injection. At the same time, the expression levels of CYP9E2 and genes related to the trehalose metabolism pathway in the surviving larvae were detected 48 h after injection. The Bio-rad CFX96™ Real-Time PCR Detection System (Bio-RAD Laboratories Inc., Hercules, CA, USA) was used for detection. PCR reaction system (10.0 μL): SYBR Green master mix (SYBR Green Premix Ex Taq, Takara, Japan) 5 μL, forward primer (10 pmol) 0.4 μL, reverse primer (10 pmol) 0.4 μL, template cDNA 1 μL, RNase Free ddH$_2$O 3.2 μL. The reaction procedure comprised pre-denaturation at 95 °C for 2 s, denaturation at 95 °C for 30 s, and annealing extension at 59c for 30 s (35 cycles). The final plotted melting curve was in a range of 65–95 °C. The relative expression was calculated via the $2^{-\Delta\Delta CT}$ method [42]. *Ribosomal Protein* L13a (*RPL13a*, GenBank ID: XM_969211) was used as an internal reference gene. The sequences of primers for qRT-PCR are shown in Table 1.

**Table 1.** Primers used for dsRNA synthesis and qRT-PCR detection.

| Application Type | Gene Name | Forward Primer (5′-3′) | Reverse Primer (5′-3′) |
|---|---|---|---|
| dsRNA synthesis | CYP9E2 | TAAGTGTGGTTTTGGGTGCG | CTGGGCTTGAATGTTAGATG |
| | T7-CYP9E2 | T7-TAAGTGTGGTTTTGGGTGCG | T7-CTGGGCTTGAATGTTAGATG |
| | dsCYP9E2 | ACGACCATCTTGCCATAA | AAACCATCACCACCTTCAT |
| | dsGFP | AAGGGCGAGGAGCTGTTCACCG | GCAGGACCATGTGATCGCGC |
| qRT-PCR | TcRPL13a | ACCATATGACCGCAGGAAAC | GGTGAATGGAGCCACTTGTT |
| | TcCYP9E2 | ACCGGTTTCAGTTTCATGGC | ACAAAGTCGGTGTTGCAGAA |
| | TcTRE-1 | AACCAAACACTCACTCATTCC | AATCCAATAAGTGTCCCAGTAG |
| | TcTRE-2 | GAAGTATCGGTTGGCTCG | GAGTGGGGTTGATTGTGC |
| | TcTRE-3 | CTTGAACGCCTTCCTCTG | CCATCCTCGTGGTCATAAA |
| | TcTRE-4 | CTACCTAAACCGCTCCCA | TGTCCAGCCAGTACCTCAG |
| | TcTRE2 | TGTTGTGCGTTTGTGCTC | GGACGGCTTATTGTTGTTTA |
| | TcTPS | GATTCGCTACATTTACGGG | GAACGGAGACACTATGAGGAC |
| | TcGS | ATTGGAGGAGTCTAGGAGTGTAC | CCGAATCGCTTTCATCAT |
| | TcGP | CCGATGGCTCCTTATGTG | GTATGCGTTTGACGTGGAT |
| | TcPFK | CTACGAAAATGTCCGAAGG | GTTGCGGTCAAAAGGTGT |
| | TcHCK1 | GAGGTATGTCTGCGAATGC | TGGAAATGTGGGTGGAAC |
| | TcPK | CAACCGACGAAAAGTATGC | TTCACCCCTTTACTACTCCC |

T7: GGATCCTAATACGACTCACTATAGG.

*2.8. Data Analysis*

SPSS 23.0 software was used for data analysis, and the mean $\pm$ standard error (SE) represents the experimental results. One-way analysis of variance (ANOVA) followed by Tukey's multiple range test was used to analyze temporal and spatial expression. Different letters on the column indicate significant differences between groups ($p < 0.05$, ANOVA). Student's t-test was used to compare the differences between the treatment group and the control. The analysis results of the t-test are indicated by *; *** stands for $p < 0.001$, ** stands for $p < 0.01$, and * stands for $p < 0.05$.

## 3. Results and Analysis

*3.1. Sequence Analysis of TcCYP9E2 Gene of T. castaneum*

Based on the transcriptome data of *T. castaneum*, the open reading frame sequence of the CYP gene related to the high-$CO_2$ stress response was cloned via RT-PCR and uploaded to the International P450 Gene Committee (Nelson's nomenclature) for naming. It was named TcCYP9E2. The predicted number of proteins encoded was 60.15, and

the theoretical isoelectric point was 8.63. There was no signal peptide; instead, there was a hydrophilic protein with transmembrane structure. The phosphorylation site was T24/S34/Y6. Multiple alignment results indicated that the gene's encoded protein had a cytochrome P450 conserved domain.

A phylogenetic tree analysis showed that *T. castaneum* TcCYP9E2 had the highest homology with the CYP9E2-like1 and CYP9E2-like2 of *Tribolium madens*. TcCYP9E2 belongs to the CYP6 family and the CYP3 group (Figure 1).

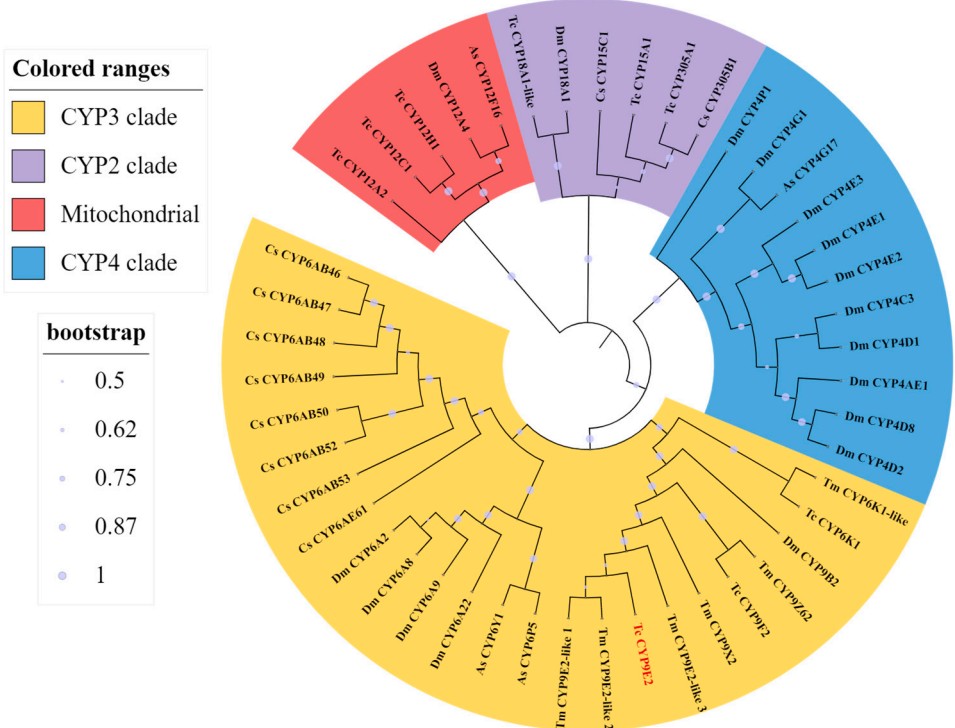

**Figure 1.** Phylogenetic tree of *TcCYP9E2* in *T. castaneum*. *Chilo suppressalis*: Cs CYP15C1 (AHW57293.1), Cs CYP305B1 (AHW57296.1), Cs CYP6AB46 (AHW57300.1), Cs CYP6AB47 (AHW57301.1), Cs CYP6AB48 (AHW57302.1), Cs CYP6AB49 (AHW57360.1), Cs CYP6AB50 (AHW57303.1), Cs CYP6AB52 (AHW57305.1), Cs CYP6AB53 (AHW57306.1), Cs CYP6AE61 (AHW57308.1); *Anopheles sinensis*: As CYP6Y1 (AIO10928.1), As CYP6P5 (AGZ03633.1), As CYP4G17 (AJF45774.1), As CYP12F16 (AKH45317.1); *Drosophila melanogaster*: Dm CYP6a2 (NP_523628.1), Dm CYP6A8 (NP_523749.2), Dm CYP18A1 (NP_523403.2), Dm CYP4G1 (NP_525031.1), Dm CYP4E2 (NP_001286196.1), Dm CYP4D1 (NP_726797.1), Dm CYP6A9 (NP_523748.2), Dm CYP4D2 (NP_001284806.1), Dm CYP4E3 (NP_523527.1), Dm CYP9B2 (NP_523646.1), Dm CYP4AE1 (NP_525044.1), Dm CYP12A4 (NP_650783.2), Dm CYP4E1 (NP_524771.1), Dm CYP4P1 (NP_524828.1), Dm CYP4C3 (NP_524598.1), Dm CYP6A22 (NP_001286431.1); *Tribolium castaneum*: Tc CYP6K1 (XP_015833657.1), Tc CYP9E2 (XP_008197432.2), Tc CYP9F2 (NP_001127706.1), Tm CYP6K1-like (XP_044265598.1), Tc CYP12C1 (XP_008199959.1), Tc CYP305A1 (XP_970235.1), Tc CYP15A1 (EFA01264.1), Tc CYP18A1-like (KYB25634.1), Tc CYP12C1 (XP_008199959.1), Tc CYP12A2 (XP_008190935.1); *Tribolium madens*: Tm CYP9E2-like×1 (XP_044272342.1), Tm CYP9E2-like×2 (XP_044262034.1), Tm CYP9E2-like×3 (XP_044253115.1), Tm CYP9X2 (AKZ17702.1), Tm CYP9Z62 (AKZ17700.1).

### 3.2. Temporal and Spatial Expression Pattern of T. castaneum TcCYP9E2

TcCYP9E2 was expressed in stages, and there was no significant difference in relative expression levels between 1–3 instars, 7–8 instars, the pupal stage, and adults 15–30A. It was mainly highly expressed in larvae of 4–6 instars and the adult 1–10A stages, and was most expressed in 6th-instar larvae, significantly more so than in other

stages, and was 121.3 times higher than the lowest expression on the fourth day of the pupal stage (Figure 2A).

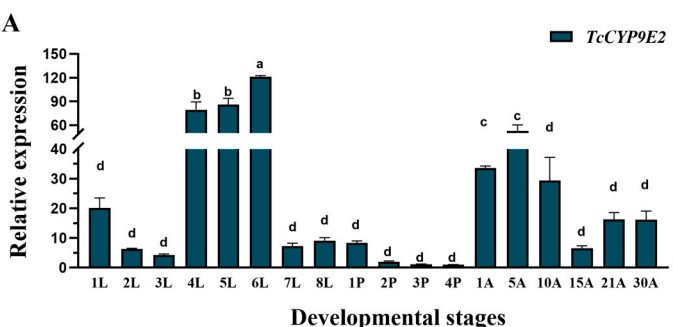

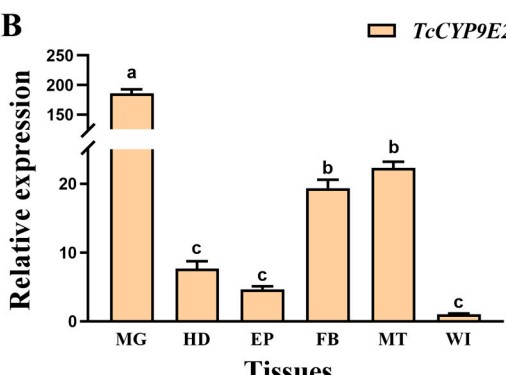

**Figure 2.** Relative expression levels of *TcCYP9E2* in different developmental stages (**A**) and tissues (**B**) of *T. castaneum.* 1L-8L: larvae 1–8 instars; 1P-4P: 1 to 4 days for pupae; 1A: on the first day of the adult; 5A: adult at 5th day; 10A: adult at 10th day; 15A: adult at 15th day; 21A: adult at 21st day; 30A: adult at 30th day; MG: midgut: HD: head: EP: epidermis: FB: fat body; MT: Malpighian tubule system; WI: wing. Three biological replicates were conducted for each development stage, and 35 test insects were collected for each replicate. Each replicate involved the dissection of 100 adult insects. (Mean ± SE; Tukey's test; different letters in the figure indicate significant differences between groups, $p < 0.05$).

The expression level of TcCYP9E2 was the lowest in the wings, and there was no significant difference in expression levels compared to the head and epidermis; the expression level of TcCYP9E2 was the highest in the midgut, 185.5 times that of the wings. Compared with the wings, TcCYP9E2 was significantly overexpressed in the Malpighian tubule system and fat bodies, 22.3 and 19.3 times as much as in the wings, respectively (Figure 2B).

### 3.3. Detection of Silencing Efficiency of T. castaneum TcCYP9E2 and Survival Rate after CO₂ Stress

The RNAi investigation of *TcCYP9E2* in 8th-instar *T. castaneum* larvae showed that, compared with the control GFP group, after 48 h of interfering with TcCYP9E2, the relative expression of its gene was significantly reduced by 70.1% (Figure 3A), which proves that the RNAi had an obvious silencing effect and subsequent experiments could be carried out.

The sensitivity changes of *T. castaneum* to $CO_2$ after dsTcYP9E2 silencing for 48 h were measured using the modified atmosphere method. The results showed that the survival rate of the dsTcCYP9E2-treated group was significantly lower than that of the dsGFP control group under different concentrations of $CO_2$ controlled atmosphere treatment ($p < 0.05$), especially with a significant decrease of 27.6% in a survival rate at 95% $CO_2$ concentration (Figure 3B).

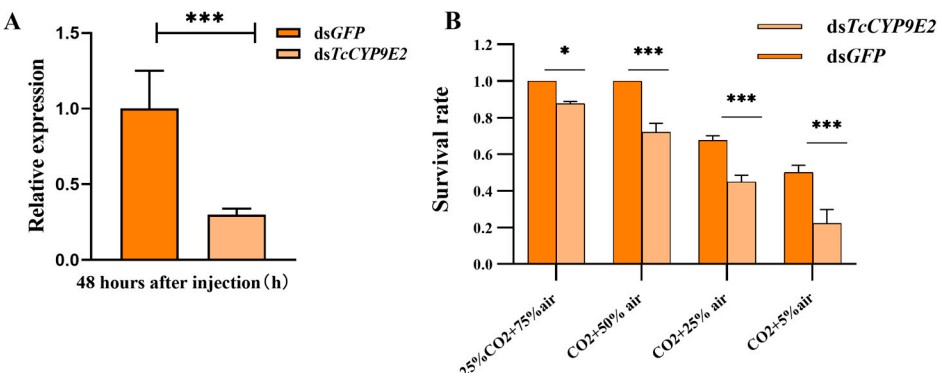

**Figure 3.** TcCYP9E2 interference efficiency (**A**) and detection of changes in its sensitivity to $CO_2$. (**B**) Values are presented as the means $\pm$ SE. ***: $p < 0.001$, *: $p < 0.05$ (independent samples *t*-test). Three biological replicates were performed on 60 *T. castaneum* larvae in each treatment.

### 3.4. Determination of Carbohydrate Content and Trehalase Activity in T. castaneum under Conditions of dsCYP9E2 Combined with 75% $CO_2$

After injecting dsRNA, the 8th instar larvae of *T. castaneum* were treated with 75% $CO_2$. The results showed that the content of glycogen and glucose decreased significantly, 0.16 times and 0.48 times the levels of the control group, respectively, and the trehalose content did not change significantly (Figure 4A).

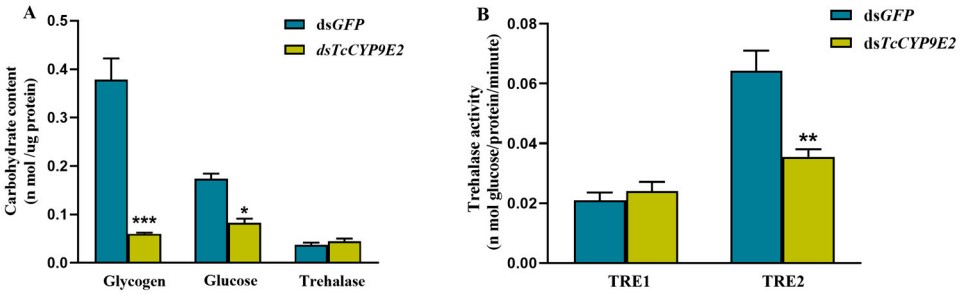

**Figure 4.** Effects of *TcCYP9E2* RNAi on the contents of glycogen (**A**) and trehalase activity (**B**) in 75% $CO_2$ condition. For each treatment, 60 larvae of *T. castaneum* were taken for 3 biological replicates. Values are presented as the means $\pm$ SE. ***: $p < 0.001$, **: $p < 0.01$, *: $p < 0.05$ (independent-samples *t*-test).

The results of trehalase activity showed that the activity of soluble trehalase in the larvae injected with dsTcCYP9E2 after 75% $CO_2$ treatment experienced no significant changes compared to the control group, while the activity of membrane-bound trehalase was significantly reduced and 0.55 times that of the control group (Figure 4B).

### 3.5. Change in T. Castaneum ATP Content under Conditions of dsCYP9E2 Combined with 75% $CO_2$

After silencing the GFP and TcCYP9E2 of *T. castaneum*, the test insects were treated with 75% $CO_2$, and the ATP content was detected. The results showed that compared to dsGFP, interference with TcCYP9E2 resulted in a significant decrease in ATP content (Figure 5).

### 3.6. The Expression Changes in Trehalose Metabolism Pathway Genes under Conditions of dsCYP9E2 Combined with 75% $CO_2$

The results showed that the expression of TRE-1 and TRE-2 in trehalose metabolism was significantly downregulated, 0.38 and 0.56 times the levels in the control group, respectively (Figure 6A). In the energy metabolism pathway, the expression of GP was significantly downregulated and 0.56 times that of the control group, while the expressions of other genes did not show significant changes (Figure 6B).

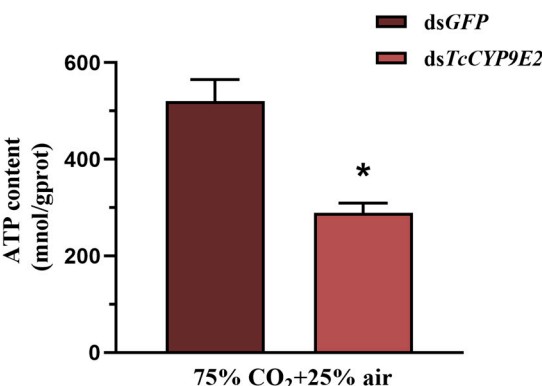

**Figure 5.** Effect of TcCYP9E2 RNAi on ATP content in *T. castaneum* in 75% $CO_2$ conditions. Values are presented as the means $\pm$ SE. *: $p < 0.05$ (independent-samples *t*-test). Three biological replicates were performed on 60 larvae of *T. castaneum* in each treatment.

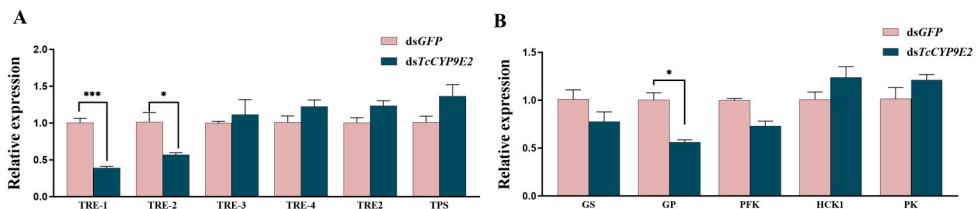

**Figure 6.** Effect of TcCYP9E2 RNAi on the expression of genes related to the trehalose metabolic pathway (**A**) and energy metabolism (**B**) in *T. castaneum* in 75% $CO_2$ conditions. TRE: trehalase; TPS: trehalose-6-phosphate synthase; GS: glycogen synthase; GP: glycogen phosphorylase; PFK: phosphofructokinase; HCK1: hexokinase; PK: pyruvate kinase. Three biological replicates were performed on 60 larvae of *T. castaneum* in each treatment. Values are presented as the means $\pm$ SE. ***: $p < 0.001$, *: $p < 0.05$ (independent-samples *t*-test).

## 4. Discussion

CYP is a type of metabolic enzyme that widely exists in all aerobic organisms. The CYP gene TcCYP9E2 obtained in this study was highly expressed in the larval stage of *T. castaneum*. Similarly, TcCYP6BQ13v2 in *T. castaneum* is highly expressed in the larval stage and is low in the pupal stage [43]. In *Spodoptera litura*, it was found that CYP6AB12 and CYP321B1 were expressed throughout the entire life cycle but were most significantly expressed during the larval stage [44,45]. The CYP gene has spatiotemporal specificity in expression, in particular CYP6AB12, CYP321B1, CYP6AB60, and CYP321A19, which are significantly expressed in the midgut and adipose tissue of *S. litura* [1]. Related studies have found that CYP311A1 is located on the endoplasmic reticulum of intestinal epithelial cells in the front of the midgut, interfering with CYP311a1 in the midgut, leading to developmental defects in microvilli in the midgut of the larva, growth stagnation, and failure to continue to develop [46]. TcCYP9E2 is highly expressed in the midgut, Malpighian tubule system, and fat bodies, which may be because CYP plays an important role in insect growth, development, and metabolism (Figure 2B). The foregut and midgut are the main sites for insect digestion and absorption [47]; the Malpighian tubule system is involved in the regulation of water and ions in insects [48]. Fat bodies can metabolize carbohydrates, lipids, and proteins and are also target tissues for hormone action [49]. It is speculated that the larval stage is the period with the strongest detoxification metabolic capacity across an insect's lifespan, and the midgut and fat bodies are the important parts of the detoxification metabolism of *T. castaneum*.

In related studies, decreases in CYP321E1 expression led to a significant increase in the sensitivity of *Plutella xylostella* to chlorobenzamide [50]. Three genes of the CYP346B subfamily (CYP346B1, CYP346B2, CYP346B3) were overexpressed in the phosphine-resistant strain of *T. castaneum*. After interfering with the expression of these three genes, the sensi-

tivity of the test insect to phosphine increased [51]. In addition, TcCYP6BQ7 in *T. castaneum* was significantly induced by the essential oil of *Artemisia argyi*, and silencing its expression increased its sensitivity to the essential oil of *A. argyi* from 49.67% to 71.67% [52]. Knocking down CYP6 family genes increased the susceptibility of *Locusta migratoria* to carbamate and pyrethrin [53]. Similarly, in our study, the expression of TcCYP9E2 in *T. castaneum* was significantly lower than in the control group after dsRNA injection (Figure 3A). Compared with the control GFP group, the mortality rate of *T. castaneum* larvae treated with 75% $CO_2$ was significantly increased. It can be concluded that TcCYP9E2 is the key factor in the response of *T. castaneum* to high concentrations of $CO_2$. The reduced expression of TcCYP9E2 increases the sensitivity of *T. castaneum* to $CO_2$ and makes it unable to adapt to high-$CO_2$ and low-$O_2$ environments; thus, it inhibits aerobic metabolism and blocks its life activities.

After the larvae injected with dsTcCYP9E2 were treated with a high-$CO_2$ environment, the balance of the carbohydrate cycle in *T. castaneum* was broken. TRE-2 is a transmembrane protein that mainly hydrolyzes trehalose in food to provide energy for muscle movement and midgut movement during feeding [54]. In combination with the high expression of TcCYP9E2 in the midgut, Malpighian tubule system, and fat bodies, it is speculated that the silencing of TcCYP9E2 impedes the metabolism of trehalose in the midgut and other parts; the insect lacks enough trehalase to catalyze the decomposition of trehalose into glucose, which limits the availability of glucose in *T. castaneum* (Figure 4). As a raw material for synthesizing glycogen, a significant reduction in glucose content not only affects glycogen content but also affects glycogen catabolism, resulting in a decrease in the expression of GP in this metabolism [55,56] (Figure 4A). At the same time, trehalose, as a natural, non-specific, cell-protective material, can form a protective film on the cell surface to maintain the biomolecular configuration, which can help a variety of organisms survive under extreme or adverse environmental pressures [57,58]. When cells are in extreme or adverse environments, they synthesize a large amount of trehalose [58]. In relevant experiments, it was found that the content of polysaccharides, soluble proteins, and fats in adult *Lasioderma serricorne* treated with high-concentration $CO_2$ was significantly reduced compared to the control [59]. It is speculated that $CO_2$-controlled atmosphere stress leads to significant energy consumption in insects, making them unable to maintain normal physiological activities and causing them to die. This experiment further detected the ATP content of *T. castaneum* and found that after silencing the CYP gene, the ATP content of *T. castaneum* decreased under 75% $CO_2$ conditions (Figure 5). ATP is the direct unit that supplies energy and is the energy source for most activities in living cells [60]. High $CO_2$ and low $O_2$ in a closed environment inhibit aerobic metabolism to a certain extent, leading to the consumption of energy in the form of ATP in the body, which is consistent with the results of the reduction in glycogen and glucose content (Figure 4A), indicating that the energy metabolism of *T. castaneum* is weakened to a certain extent in this case, but the potential functions of the glucose metabolism pathway are different and more complex, which necessitates further in-depth research. The above results suggest that CYP may affect the gas resistance of *T. castaneum* through the synthesis or decomposition routes of the glucose metabolism pathway.

## 5. Conclusions

In this paper, dsRNA combined with high-$CO_2$ controlled atmosphere treatment was used to silence the CYP gene TcCYP9E2, and the control effect of this method on storage pests was explored through the determination of the content of related substances, enzyme activity, and gene expression level changes. The TcCYP9E2 of *T. castaneum* has spatiotemporal expression and spatial expression specificity. It is speculated that the larval stage is the strongest period of detoxification metabolism across an insect's lifespan, and the midgut and fat body are the important parts of the detoxification metabolism of *T. castaneum*. After treating RNAi-silenced-TcCYP9E2 *T. castaneum* with different concentrations of $CO_2$, its survival rate decreased with the increase in $CO_2$ concentration. It can be concluded

that the CYP gene TcCYP9E2 is the key factor for *T. castaneum* to respond to high $CO_2$ concentrations. Reduction in the expression of TcCYP9E2 enhances the sensitivity of *T. castaneum* to $CO_2$, so the CPY gene TcCYP9E2 plays a metabolic role in the process of *T. castaneum* coping with high-$CO_2$ stress. Compared with the dsGFP control, the glycogen content, ATP content, and membrane-bound trehalase activity of *T. castaneum* subjected to TcCYP9E2 inhibition combined with a 75% $CO_2$ environment significantly decreased. The energy metabolism of *T. castaneum* was weakened to a certain extent, but the potential function of the carbohydrate metabolism pathway is different and more complex, which necessitates further in-depth research, indicating that CPY may affect the gas resistance of *T. castaneum* through the synthesis or decomposition routes of the carbohydrate metabolism pathway.

**Author Contributions:** Y.-F.Z., M.Z. and Y.-Y.W.: Methodology, Investigation, Formal analysis, Writing—Review and Editing. X.-Y.J. and P.Z.: Formal analysis, Writing—Review and Editing. K.-K.X., B.T. and C.L.: Conceptualization, Supervision, Writing—Review and Editing, Project administration. All authors have read and agreed to the published version of the manuscript.

**Funding:** This research was financially supported by the National Natural Science Foundation of China (Grant No. 31960542), the Discipline and Master's Site Construction Project of Guiyang University by Guiyang City Financial Support Guiyang University [2022-xk12] and the Program for Natural Science Research in Guizhou Education Department (QJJ [2023]024), the Special Project for Science and Technology Development of Local (Guizhou) under the Guidance of the Central Government (QKZYD [2022]4013), and Guiyang University Supports Graduate Research Project (GYU-YJS(2021)-64).

**Institutional Review Board Statement:** Not applicable.

**Informed Consent Statement:** Not applicable.

**Data Availability Statement:** Not applicable.

**Conflicts of Interest:** The authors declare no conflict of interest.

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
