# Peer review of "Characterization of the TcCYPE2 Gene and Its Role in Regulating Trehalose Metabolism in Response to High CO2 Stress"

_agronomy, doi:10.3390/agronomy13092263_

Round 1

Reviewer 1 Report

The manuscript entitled "TcCYP9E2 cooperates with trehalose metabolism in response to high-CO2 stress" the expression of Cytochrome P450 monooxygenase (CYP) in stored grain pest (Tribolium castaneum) upon exposure to carbon dioxide stress. The findings of the research have significant importance for the insect pest management. This low oxygen in grain storage system will surely assist policy makers to develop effective pest management programs. The manuscript is very well presented, there are some minor mistakes that have been highlighted in annotated file for improvement.

There are minor issues related to English Language (highlighted in the text)

Author Response

Please see annex.

Reviewer 2 Report

This work identified a CYP9E2 of Tribolium castaneum in response to high-CO2 stress, and used RNAi to reveal the molecular and physiological mechanism of CYP9E2 in the adaptation of T. castaneum to high-CO2 stress. The structure of manuscript is fine and the findings are interesting. It could be useful for the sustained and effective prevention and control of the stored grain pests.

There is only one concern: From the result of Figure 3, it could be inferred that 95% CO2 treatment resulted in the lowest survival rate. It would be better to reason why the subsequent experiments are all based on the treatment of 75% CO2.

Other issues:

1. Almost all the scientific name “T. castaneum” is not in italics.

2. The caption of Y axis “Relative Expression” of Figures 2, 3 and 6 can be changed to “Relative expression”.

Author Response

Please see annex.

Reviewer 3 Report

the title of the study seems to be wrong, how it cooperates to trehalose? the author only perfomed qPCR analysis when the specific gene was knockdown, why only the one gene was knockdown?

The introduction lack flow, should be re-written with a flow that a reader can understand from the broader to the current study hypothesis. The author should write the introduction in a way that can explain, why pesticides are not good? how it harmful for the environment? what are the alternatives ways? For example: Essential oils, botanicals and the use of micro and nano-capsules using for the control of different pests.

As the author is specific about stored product pests, use of EO and active compounds of EO are used for the control of several stored product pests, that caused energy homeostasis disruption. the author should incorporate. here are few articles, the author may extract data: 10.1016/j.ecoenv.2022.114304, 10.3390/cells12040669 , 

Line 99, "it will used for new nucleic acid preparation" what is nucleic acid preparation???

Line 96, it was speculated" on the basis of what?? how it is important for metabolism???

21 genes were significantly differentially expressed, according to line 94, then how and why select only 1?

L 106 what is mean by hangzhou normal university castaneum??

L128, here and elsewhere, the name should be italic.

write again section 2.3 with correct english

L142, the RNA was checked on 1.5 agarose gel, can the author provide those gel pictures in supplementary data?

Rewrite 2.4 with correct english.

There is no primer for the internal control being used for the qPCR.

The author should re-write the whole methodology section again. so many flaws and irrelevant information in every section. the author put the primers in two tables, with a wrong titles, this should move to supplemenatry data. The methodology is not in accordance with the relevant protocol, which rise questions on the integrity of the experiments. 

please mention every model, CAS number and information about Reagents being used for experiments.

Section 2.8, the data analysis, pleae re-write.

It's very difficult for the reader to read the results, as it is wrongly written. please re-write all section, with the assistance of any senior in a lab, or revise it by the expert or copy editor. 

so many irrelevant citation.

I would say, rewrite the manuscript in a good english, polish it for english language, scientific soundness and the re-submit it. I am unable at the moment to understand this manuscript. 

please read more papers and with the help of any senior or any english editing service, improve the english of the manuscript. at current, it extremely difficult to understand. 

Author Response

Please see annex.

Round 2

Reviewer 3 Report

the authors did well and addressed all the concern.